# A Study of the Degradation of LEV by Transparent PVA/NCD-TiO$_2$ Nanocomposite Films with Enhanced Visible-Light Photocatalytic Activity

**Anhua Jiang [1], Xinwen Huang [1,*], Geshan Zhang [2,3,*] and Wanquan Yang [4]**

[1] College of Environment, Zhejiang University of Technology, Hangzhou 310014, China
[2] College of Chemical Engineering, Zhejiang University of Technology, Hangzhou 310014, China
[3] School of Chemistry and Environmental Engineering, Wuhan Institute of Technology, Wuhan 430205, China
[4] Powerchina Huadong Engineering Corporation Limited, Hangzhou 311122, China
[*] Correspondence: xwhuang@zjut.edu.cn (X.H.); zhanggs@zjut.edu.cn (G.Z.); Tel.: +86-571-88320412 (G.Z.)

**Abstract:** In recent years, antibiotics (such as levofloxacin (LEV)) have been detected widely in the environment. Semiconductor photocatalysis has been recognized as a promising technology for removing pollutants in the environment. In this work, nitrogen and carbon codoped titanium dioxide nano-catalyst (NCD-TiO$_2$) was immobilized in polyvinyl alcohol (PVA) matrix to form PVA/NCD-TiO$_2$ films through solution casting and thermal treatment, which exhibited good photocatalytic efficiency for LEV degradation. The results showed that about 42% LEV can be degraded after 2 h in the presence of PVA/NCD-TiO$_2$ nanocomposite film (the weight ratio of NCD-TiO$_2$ to PVA is 8% and thermal treatment is 120 °C) under visible light. Moreover, possible pathways of photocatalytic degradation of LEV according to the detected intermediates are proposed, which provide insight into the degradation mechanism of LEV by using PVA/NCD-TiO$_2$ photocatalytic films. Finally, the synthesized PVA/NCD-TiO$_2$ films exhibited excellent reusability and stability in photocatalysis. This work provides fundamental support for the design of a high-stability, excellent photocatalyst for practical application.

**Keywords:** levofloxacin; polyvinyl alcohol; visible-light photocatalysis; titanium dioxide; water treatment

## 1. Introduction

Levofloxacin (LEV) is a synthetic fluoroquinolone antibiotic that has been widely used for treatment of severe bacterial infections [1,2]. However, the antibiotics cannot be absolutely metabolized by humans or animals and thus can be released into environment as drug-active forms through the excretory system [3,4]. Municipal and hospital wastewater are the main sources of these antibiotics in natural waters [5], which can facilitate the dissemination of antibiotic resistance by any plasmid-mediated transformability [6–8]. It was found that the abundance and transfer of bacteria with antibiotic resistance genes was concomitant with the presence of antibiotic residues and consequently contributed to the spread of antibiotic-resistance genes [9]. In addition, the coexistence of antibiotics and antibiotic-resistance genes can amplify existing resistance genes and create new resistance genes or genomic assemblages [10], which threaten the safety of the environment and human health. Therefore, seeking an effective approach to remove antibiotics, such as LEV, in water is of great importance.

Heterogeneous photocatalytic processes utilizing solar energy have been found to be a promising way to solve environmental crises, as they can convert environmental contaminants into harmless compounds [11,12]. Thus far, a large number of semiconductors, such as BiOIO$_3$ [13], ZnO [14], Ag/g-C$_3$N$_4$-Ag-Ag$_3$PO$_4$ [15], Ag$_2$O/Bi$_{12}$GeO$_{20}$ [16], and RGO/In$_2$TiO$_5$ [17], have been found to be active photocatalysts for the photodegradation of various organic contaminants. Among them, TiO$_2$ has been one of the most promising

photocatalysts due to its low cost, nontoxicity, and long-term stability [7,18]. Since the band gap of bulk $TiO_2$ is wide (3.0 eV for rutile phase and 3.2 eV for anatase phase), normal $TiO_2$ can only be activated by ultraviolet light (UV), which is only a small portion (<10%) of the solar energy. Therefore, the development of visible-light active photocatalysts is of great significance for the utilization of solar energy in detoxifying waters [19,20].

$TiO_2$ photocatalyst has excellent catalytic performance for the treatment of industrial dying wastewater [21], pharmaceutical wastewater [22,23], and other wastewaters [24]. Khalid et al. synthesized graphene–$TiO_2$ composites with high efficiency for the degradation of methyl orange under visible light [25]. Kaur et al. prepared Cu-doped $TiO_2$ as a photocatalyst for the eradication of an antibiotic (ofloxacin) from an aqueous phase under visible illuminations, which showed high photocatalytic activity [26].

Another disadvantage of $TiO_2$ powders is the high cost of subsequent separation. In order to overcome this shortage, many materials were applied for immobilization of $TiO_2$ including zeolites [27], glass spheres [28], magnetic materials [29], and polymers [30], which can provide the catalyst relatively high quantum utilization efficiency and ease of posttreatment [31]. Tennakone et al. firstly reported the use of polymer support materials in $TiO_2$ photocatalysis [32]. Chu et al. investigated the effects of pH, molecular weight, and grade of hydrolysis of poly (vinyl alcohol) on the performance of PVA (slot die coating)-supported $TiO_2$ [33]. In our previous work, we had developed a facile method for preparation of N-C-doped activity visible-light-driven (NCD-$TiO_2$) nanoparticles, which had outstanding visible-light photocatalytic activity for ciprofloxacin (CIP) and LEV removal [33] and that can be combined with PVA support for a wider and convenient application. To the best of the authors' knowledge, however, there is no report on PVA-supported $TiO_2$ with visible-light activity, especially for the removal of LEV.

In this paper, we describe a simple route to prepare a crosslinked PVA/NCD-$TiO_2$ hybrid system with high photocatalytic efficiency for the degradation of levofloxacin in aqueous phase for multicycle use which combines solution-casting and heat treatment. The structural properties and morphology of the synthesized PVA/NCD-$TiO_2$ film were studied in detail according to various spectroscopic and analytical techniques (SEM, TEM, XRD, FTIR). The effects of temperature in heat treatment, catalyst amount, and different molds on the photocatalytic performance were explored. Based on the intermediates identified by LC-MS, a photocatalytic degradation pathway of levofloxacin is also proposed.

## 2. Results and Discussion

### 2.1. Physico-Chemical Characteristics of Synthesized Titania Catalysts

2.1.1. Immobilization of NCD-$TiO_2$ Nanoparticles in PVA Matrix

In the comparison of the SEM images of films, the pure PVA film was relatively smooth, while all the PVA/NCDPVA/NCD-$TiO_2$ films became rougher due to the addition of NCD-$TiO_2$ as well as the agglomeration of the nanoparticles (see Figure 1 and Figure S1 in Supplementary Materials). The surface morphology of the PVA/NCDPVA/NCD-$TiO_2$ showed many NCD-$TiO_2$ aggregates or chunks that were randomly distributed on the surface of the film, indicating the poor dispersion of NCD-$TiO_2$ nanoparticles in the PVA matrix. The difference between PVA/NCDPVA/NCD-$TiO_2$ films was not significant. However, Figure 1c,d show the SEM images of the PVA/NCDPVA/NCD-$TiO_2$ calcined at 120 °C and 160 °C, respectively, suggesting a filamentous structure. This structure was probably caused by adsorption or embedding of NCD-$TiO_2$ on the surface layer of PVA as well as specific dynamic action that can be attributed to the chemical bonds formed between $TiO_2$ nanoparticles and the PVA matrix [34]. Sample 8 wt%-120 had more intensive silkiness (Figure 1c), which indicates that higher temperature (i.e., 160 °C) may reduce filamentous structures. Figure 1e,f exhibit the HR-TEM images of the NCD-$TiO_2$, which have clear lattice fringes corresponding to (101) crystal planes of anatase $TiO_2$, hinting at its high crystallinity and potential for high catalytic activity [34].

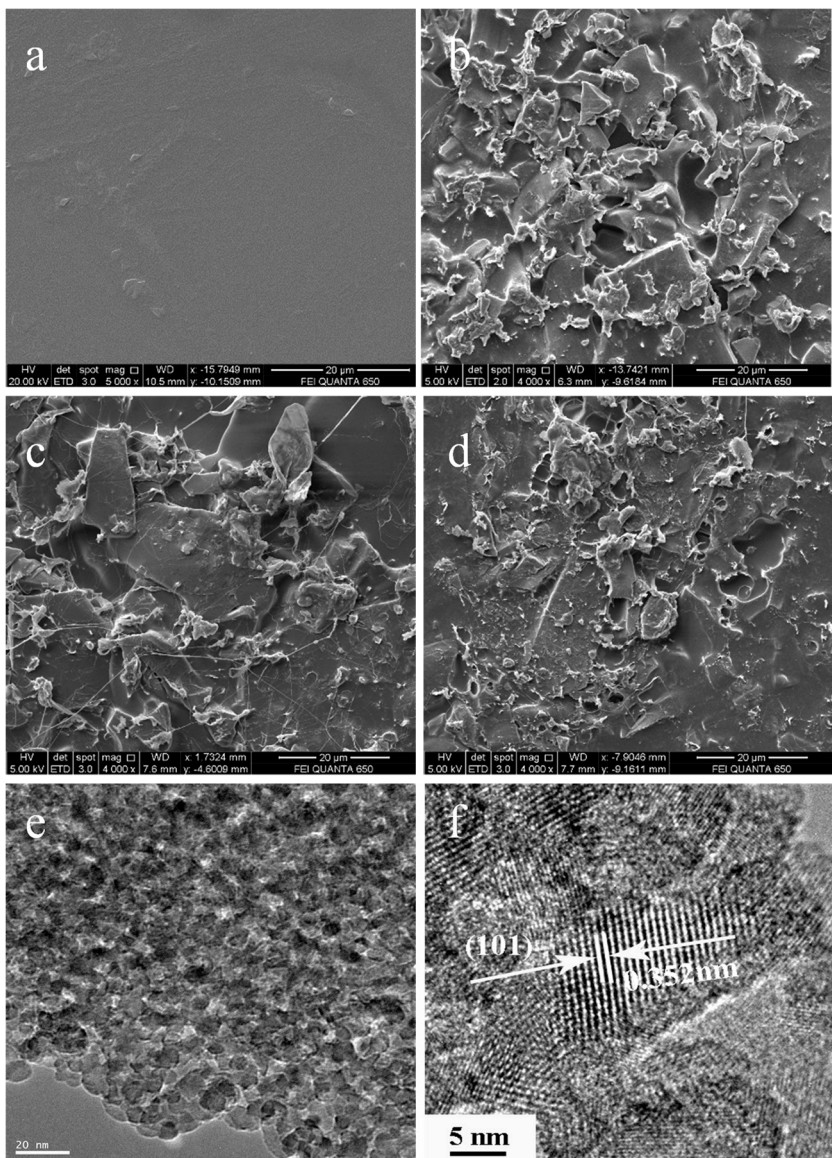

**Figure 1.** SEM images of the surfaces of (**a**) pure PVA, (**b**) 8 wt%-0-PVA/NCD-TiO$_2$, (**c**) 8 wt%-120-PVA/NCD-TiO$_2$, and (**d**) 8 wt%-160-PVA/NCD-TiO$_2$; TEM images of (**e**,**f**) the NCD-TiO$_2$ nanoparticle.

### 2.1.2. X-ray Diffraction Analysis

The X-ray diffraction technique was performed to identify the crystal structure and phase composition of PVA, 8 wt%-0-PVA/TiO$_2$, 8 wt%-120-PVA/TiO$_2$, and NCD-TiO$_2$. According to JCPDS card no. 99-0008, all synthesized NCD-TiO$_2$ photocatalysts displayed the characteristic peaks of anatase TiO$_2$. In Figure 2, the peak identified at 2θ ≈ 20° should be assigned to the response of PVA [21]. The XRD patterns of 8 wt%-120-PVA/TiO$_2$ are very similar to those of 8 wt%-0-PVA/TiO$_2$, both of which have diffraction peaks at 19.55° (i.e., PVA) and 25.35° (anatase TiO$_2$ (101)) [34]. Moreover, the crystal form of NCD-TiO$_2$ remained unchanged during preparation. The Raman spectra (see Figure S2 in Supplementary Materials) also support the conclusions.

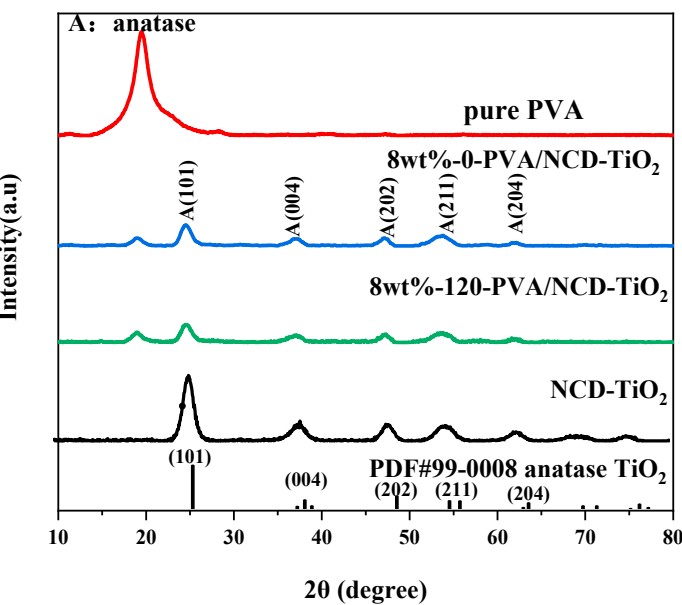

**Figure 2.** XRD spectra of pure PVA, 8 wt%-0-PVA/NCD-TiO$_2$, and 8 wt%-120-PVA/NCD-TiO$_2$.

### 2.1.3. FTIR Spectra of the Fabricated Membrane

To confirm the immobilization of NCD-TiO$_2$ in PVA, ATR/FT-IR spectroscopy was performed in this work for the structure characterization. In Figure 3, only 3 obvious characteristic absorption peaks of each band are noted in the IR spectrum of the NCD-TiO$_2$ sample, where the characteristic peaks appear around 1627 cm$^{-1}$ is the characteristic absorption peak of H–O–H bond, which should result from the absorption of moisture in air; the most dominant peak in the whole spectrum appears at the band below 800 cm$^{-1}$ which mainly contains the characteristic absorption peak of the Ti–O bond and Ti–O–Ti bond [35].

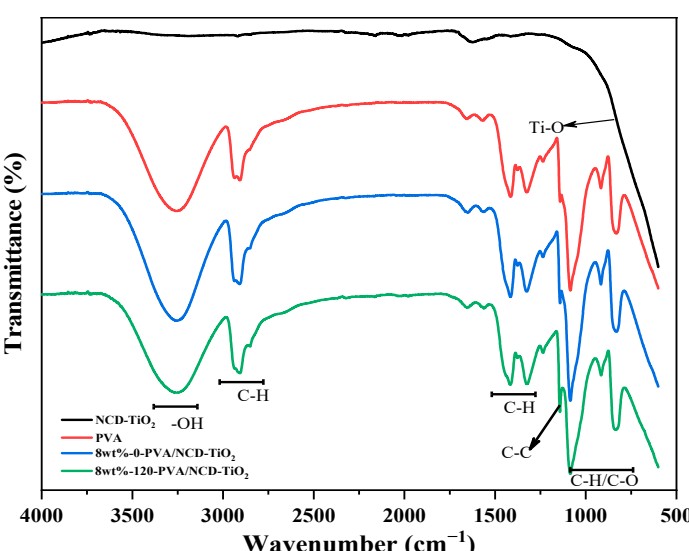

**Figure 3.** ATR/FTIR spectra of various samples.

The IR spectrum of PVA is more complex, and 9 obvious characteristic absorption peaks can be observed. Among them, the stronger characteristic peak appearing around 3257 cm$^{-1}$ should be the stretching vibration peak of -OH, which had a broad peak shape because intermolecular or intramolecular hydroxyl groups would associate with other structures and thus shift the characteristic peak. The characteristic peaks around 2932 cm$^{-1}$ and 2900 cm$^{-1}$ mainly include symmetric–antisymmetric stretching vibration

of methyl/methylene groups [36]. Two faint characteristic peaks of the sample appear around $1600 \text{ cm}^{-1}$, while the one at the larger wavelength is related to water molecules and the other one might originate from the stretching vibration of C-C. The absorption peaks at $1415 \text{ cm}^{-1}$ and $1324 \text{ cm}^{-1}$ mainly originate from the antisymmetric stretching vibration of methyl/methylene groups [37]. A distinct shoulder peak at $1142 \text{ cm}^{-1}$, which is more critical and related to the crystallinity of PVA, is the C-C stretching vibration peak associated with the stretching vibration of C-O; that is, the symmetric stretching vibration of O-C-C [36,38]. The absorption peak at $1093 \text{ cm}^{-1}$ is the C-O stretching vibration peak associated with the stretching vibration of C-C; that is, the antisymmetric stretching vibration peak of O-C-C [38]. The characteristic peaks present at $916 \text{ cm}^{-1}$, $849 \text{ cm}^{-1}$ and smaller wavelengths are mainly associated with the stretching vibration of the aliphatic hydrocarbon backbone (mainly including C-H and C-O, etc.) [38,39].

As can be seen from the infrared spectrum of 8 wt%-0-PVA/NCD-TiO$_2$ without thermal treatment, the shape and position of the characteristic peaks of the overall infrared spectrum of PVA does not change significantly after the modification, which indicates that probably only a simple physical binding state exists between the two ingredients and that the TiO$_2$ particles are simply embedded in the PVA support without forming new chemical bonds. Although there is no obvious chemical cross-linking between PVA and NCD-TiO$_2$, after fitting the infrared profiles of PVA and 8 wt%-0-PVA/NCD-TiO$_2$, it is apparent that the relative intensity of the band below $800 \text{ cm}^{-1}$ increases from 16.75% (PVA) to 29.81% (8 wt%-0-PVA/NCD-TiO$_2$) after modification, which should result from the introduction of a large amount of Ti-O. Meanwhile, the PVA crystallinity is calculated through

$$X = \frac{A_{1142}}{A_{1142} + A_{1093}} \times 100\%$$

where X is the PVA crystallinity while $A_{1142}$ and $A_{1093}$ are the absorption peak areas at $1142 \text{ cm}^{-1}$ versus $1093 \text{ cm}^{-1}$, respectively [40].

The results show that the PVA crystallinity decreased from 0.1216 (PVA) to 0.1185 (8 wt%-0-PVA/NCD-TiO$_2$) after modification, indicating that the introduction of NCD-TiO$_2$ changed the physical properties of PVA, i.e., there was improvement in the ductility and a decrease in the permeability [40,41].

New characteristic peaks were not observed in the IR spectrum of 8 wt%-120-PVA/NCD-TiO$_2$; however, the changes in shape and intensity were apparent when compared to the spectrum of 8 wt%-0-PVA/NCD-TiO$_2$ (Figure 3). In order to better explore the internal molecular structure changes of materials under different heating treatment, the infrared spectra of the samples were investigated by applying the heating temperature as the variable to form a data set for two-dimensional correlation spectral analysis. Two dimensional correlation analysis provided two different maps (Figure 4). The synchronous spectrum shows the homology of the spectrum signal (the change direction and change rate of functional groups), and the asynchronous correlation spectrum correlates the information with the sequence of events (i.e., the change sequence of thermal decomposition/synthesis of functional groups). In the synchronous spectrum (Figure 4a), five distinct characteristic absorption peaks can be observed: $3240 \text{ cm}^{-1}$, $2910 \text{ cm}^{-1}$, $1410 \text{ cm}^{-1}$, $1080 \text{ cm}^{-1}$, and $640 \text{ cm}^{-1}$. All the cross peaks show positive signals, indicating that the five functional groups change in the same direction and increase significantly during the heating treatment. Combined with the automatic peak intensity (i.e., the depth of color of the red peak on the diagonal), this suggests that the change rate of hydroxyl and Ti-O/C-O is high. According to the asynchronous spectrum (Figure 4b), the change order of the five functional groups is $2910 \text{ cm}^{-1} > 1410 \text{ cm}^{-1} > 1080 \text{ cm}^{-1} > 3240 \text{ cm}^{-1} > 640 \text{ cm}^{-1}$, which indicates that the organic framework of PVA is sensitive to temperature and that the hydrogen bonding between NCD-TiO$_2$ and PVA would be affected by the change of the crystalline structure of PVA [42]. Compared with the sample without heating treatment, the oxygen-containing functional groups around $1080 \text{ cm}^{-1}$ increase significantly with the temperature, indicating that the interaction between the oxygen-containing functional groups on the organic chain of PVA and NCD-TiO$_2$ increases during heating treatment. The drastic

change of 3240 cm$^{-1}$ (hydrogen bond) further supports the conclusion. Moreover, a binding of the Ti–O–C bond may be formed, which can fix NCD-TiO$_2$ nanoparticles effectively in PVA matrix. The increase in the relative strength of 640 cm$^{-1}$ indicates that the physical binding between NCD-TiO$_2$ and PVA also increases with the temperature. The increase in the content of hydroxyl and C-H can facilitate PVA to be a transparent support for NCD-TiO$_2$, which would effectively capture holes generated by light and thus improve the photodegradation rate of pollutants [43,44]. It is worth noting that the C–O bond (around 1142 cm$^{-1}$) becomes more obvious after the increasing temperature of treatment, indicating that heating can improve the crystallinity of the material, which is related to the strong interaction between NCD-TiO$_2$ and PVA [40].

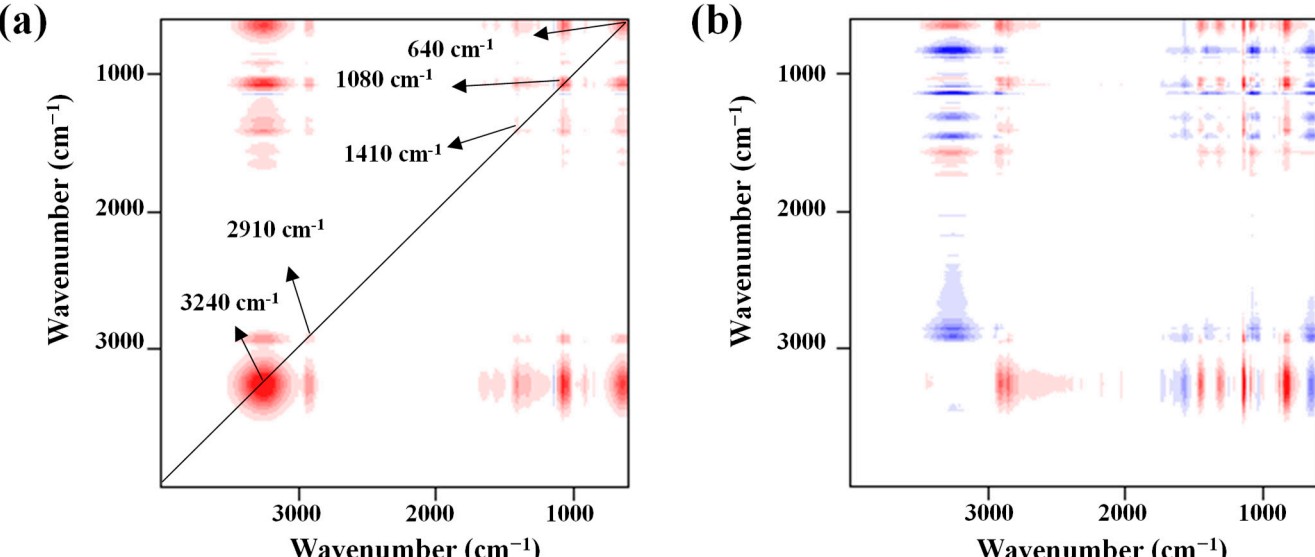

**Figure 4.** 2DCOS analysis diagram of PVA/NCD-TiO$_2$. (**a**) Synchronous spectral analysis and (**b**) asynchronous spectral analysis.

### 2.2. Catalytic Activities of the PVA/NCD-TiO$_2$ Membrane

#### 2.2.1. Effect of Heating-Treatment Temperature on Photocatalytic Activity

The photocatalytic data of NCD-TiO$_2$ can be found in our previous work [33]. The UV–vis absorption spectrum of PVA/NCD-TiO$_2$ (see Figure S3 in Supplementary Materials) shows that the synthesized membrane still has visible-light activity hinting at its photocatalytic performance. When the PVA/NCD-TiO$_2$ membrane was applied in the photocatalytic experiment, the stability of photocatalysis could be observed within four cycles, as shown in Figure 5. In addition, the direct photolysis of phenol and LEV under visible light was found to be negligible. The kinetics of the degradation of phenol (a typical organic contaminant in various waters) and LEV were further fitted with a pseudo first-order rate equation

$$\ln(\frac{C_0}{C_t}) = kt$$

where, $C_0$ is the initial concentration of contaminant (i.e., phenol or LEV), $C_t$ is the concentration of contaminant at time t, and k is the corresponding rate constant.

Figure 5 shows the degradation of phenol and LEV in four cycles photocatalyzed with PVA/NCD-TiO$_2$ hybrid films treated at different temperatures. It can be seen that the PVA/NCD-TiO$_2$ film without thermal treatment (i.e., 8 wt%-0-PVA/NCD-TiO$_2$) has better initial photocatalytic activities for degradation of both phenol and LEV. However, its later photocatalytic activity shows a significant decrease, probably because of the lack of interaction between TiO$_2$ and PVA without thermal treatment, which would result in the loss of the catalyst during cyclic usage. The catalyst treated at 80 °C (i.e., 8 wt%-80-PVA/NCD-TiO$_2$) has a close performance to 8 wt%-0-PVA/NCD-TiO$_2$, indicating that its

temperature of heating treatment did not reach the requirement for the reaction to form the interaction between $TiO_2$ and PVA. The catalytic efficiency of 8 wt%-120-PVA/NCD-$TiO_2$ only showed slight decrease within four cycles, whereas the 8 wt%-160-PVA/NCD-$TiO_2$ film showed a lower catalytic activity, suggesting that 120 °C is the optimized temperature for thermal treatment. In addition, the leaching of $TiO_2$ after photocatalysis was measured through ICP analysis to be within 10% for 8 wt%-120-PVA/NCD-$TiO_2$. In summary, the heating treatment requires suitable temperature, as a too-high or too-low temperature would not form hydrogen bonds between the O-H in PVA and the terminal hydroxyl groups on the NCD-$TiO_2$ surface, which would eventually affect the performance of catalyst during cyclic usage.

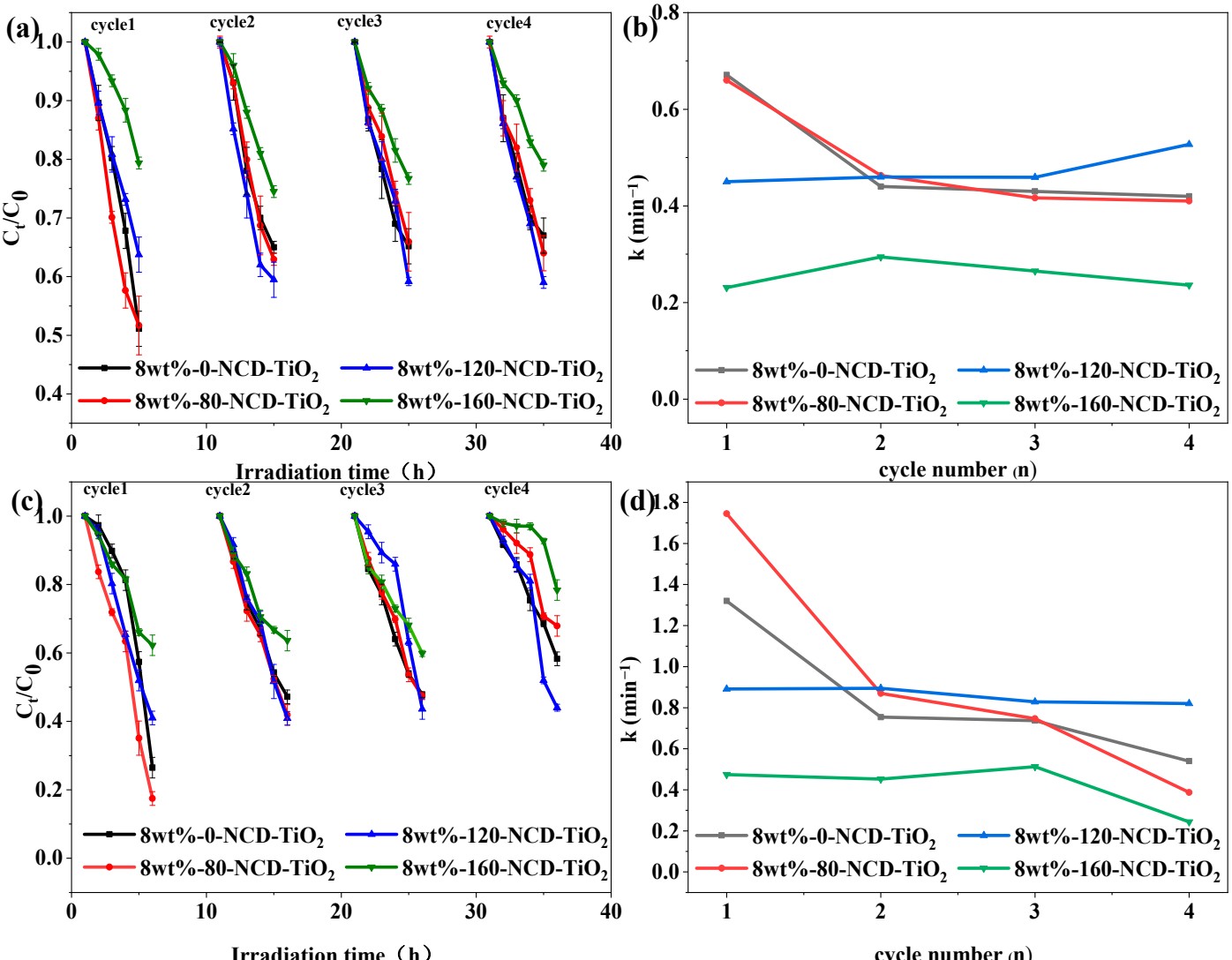

**Figure 5.** (**a**) Degradation of phenol and (**b**) the reaction rate constant k for the photocatalytic degradation of phenol during four cycles in the presence of the PVA/NCD-$TiO_2$ hybrid films treated at different temperatures; (**c**) degradation of LEV and (**d**) the reaction rate constant k for the photocatalytic degradation of LEV during four cycles in the presence of the PVA/NCD-$TiO_2$ hybrid films treated at different temperatures.

### 2.2.2. Effect of NCD-$TiO_2$ Loading on the Performance of Hybrid PVA/NCD-$TiO_2$ Films

The contaminant removal rate with nanohybrid films with various concentrations of NCD-$TiO_2$ nanoparticle was investigated, as shown in Figure 6. All PVA/NCD-$TiO_2$ films were thermally treated at 120 °C to ensure the stability of the catalytic films. It can

be observed that the photocatalytic activities of the films were significantly improved, as the weight ratio of NCD-TiO$_2$ to PVA increased from 4 wt% to 20 wt%. The enhancement in photocatalytic activity can be attributed to a higher amount of active NCD-TiO$_2$ and the hydrophilicity behavior of the NCD-TiO$_2$ nanoparticle [45], which can increase the wettability of the membrane surface. The difference between the sample 4 wt%-120-NCD-TiO$_2$ and 8 wt%-120NCD-TiO$_2$ is very apparent (2.25 times in k for phenol and 2.41 times for LEV). The rising trend of photocatalytic activity decreased with an increase in NCD-TiO$_2$ content, which may be attributed to the fact that the increasing concentration of NCD-TiO$_2$ nanoparticles led to decreased surface activity and inhibited the contact of pollutants with nanoparticles. A too-high concentration of NCD-TiO$_2$ has been reported to result in more severe aggregation, which greatly reduces the surface area and is unfavorable for pollutant photocatalytic degradation [46].

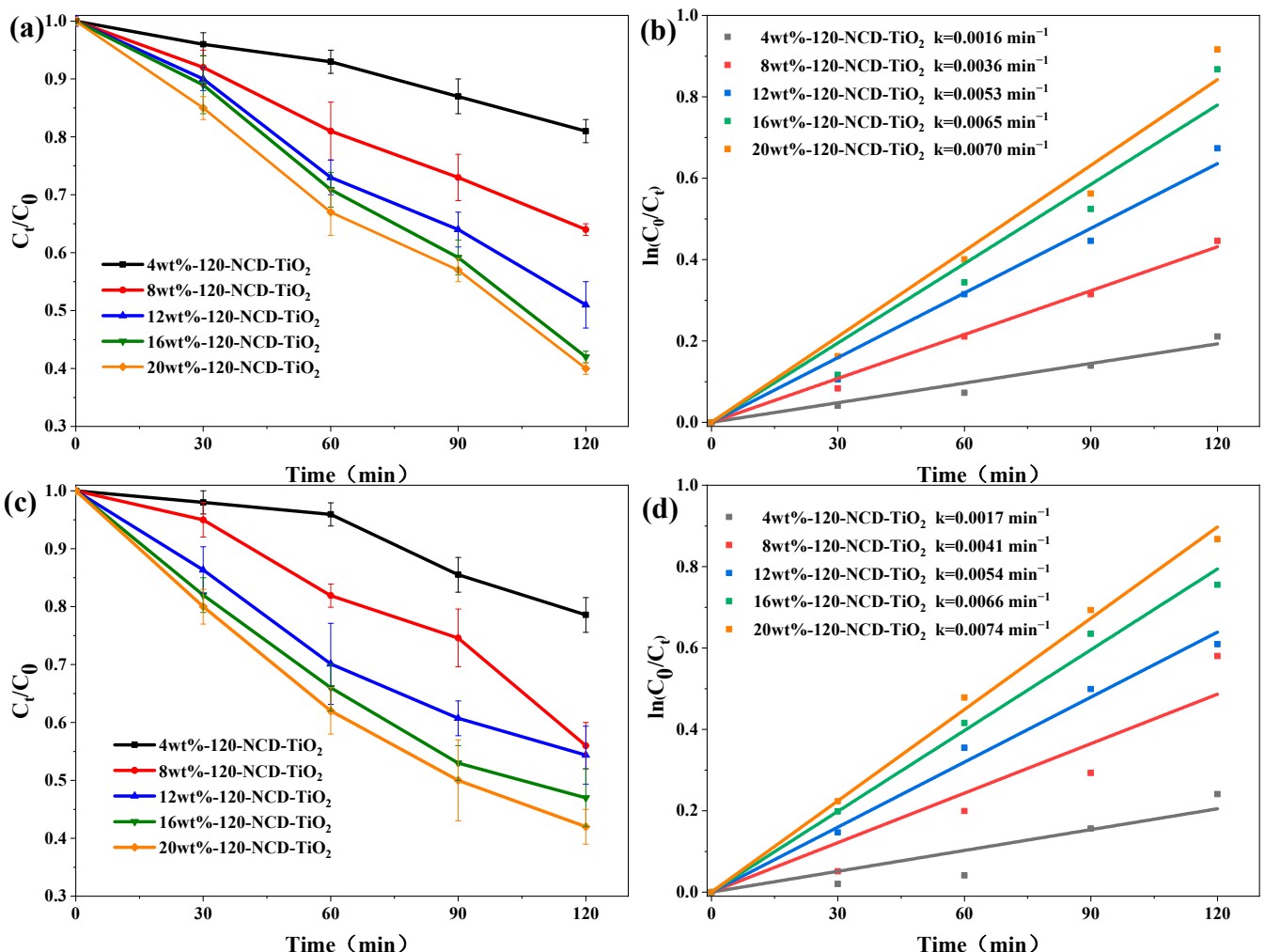

**Figure 6.** Photocatalytic activities of PVA/NCD-TiO$_2$ nanocomposite films with various concentrations of NCD-TiO$_2$ on the degradation of phenol (**a,b**) and LEV (**c,d**) under visible-light irradiation.

#### 2.2.3. Effect of Different Molds of PVA/NCD-TiO$_2$ Films on Photocatalytic Performance

The effect of thickness of PVA/NCD-TiO$_2$ films on its photocatalytic activity was also examined. The films with different thickness were prepared by static casting the same PVA/NCD-TiO$_2$ solution with different molds (the inset of Figure 7). The effect of 3 thicknesses (i.e., 1 cm, 0.75 cm, 0.5 cm) on the catalytic activity was checked, as displayed in Figure 7. The experimental results show that the PVA/NCD-TiO$_2$ film with the thickness of 0.5 cm had the best photocatalytic activity, suggesting a thicker film would not benefit

the catalytic efficiency. The thinner but larger film would result in better dispersion of NCD-TiO$_2$ and a larger active area for visible-light photocatalytic illumination, which can eventually improve the photocatalytic efficiency. Hence, the film should be as thin as possible (e.g., 0.5 cm) when the strength of the film is guaranteed.

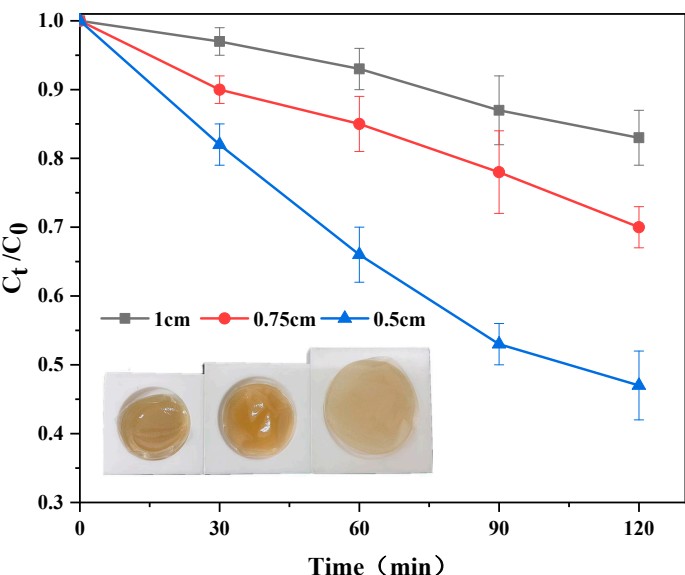

**Figure 7.** Photocatalytic activity of the PVA/NCD-TiO$_2$ nanocomposite films fabricated with different thicknesses for LEV removal under visible light.

### 2.2.4. Applications of PVA/NCD-TiO$_2$ in Natural Water

To verify the effect of PVA/NCD-TiO$_2$ photocatalytic membranes in practical water applications, the removal experiments of COD$_{Cr}$ from natural water bodies were investigated. The results shows that the PVA/NCD-TiO$_2$ photocatalytic membranes (8 wt%-120-PVA/TiO$_2$) showed different removal efficiency for different waters, while the removal efficiency for the three water samples were all above 60% as shown in Table 1. That the PVA/NCD-TiO$_2$ photocatalytic membranes showed different degradation efficiencies in different waters should be attributed to the differences in pH, COD$_{cr}$ concentration, and inorganic salt in the water.

**Table 1.** Photocatalytic activity of 8 wt%-120-PVA/TiO$_2$ in natural water.

| Real Water | Initial COD$_{Cr}$ (mg/L) | 2 h COD$_{Cr}$ (mg/L) | pH | Removal Rate (%) |
|---|---|---|---|---|
| West Lake | 25.6 ± 0.3 | 6.7 ± 0.3 | 6.42 ± 0.2 | 73.8 |
| Shangtang River | 20.1 ± 0.2 | 7.1 ± 0.2 | 6.11 ± 0.1 | 64.7 |
| Guxin River | 23.2 ± 0.2 | 7.3 ± 0.3 | 6.33 ± 0.1 | 68.5 |

On the basis of the above discussion and the results from our previous reported work [34], the mechanism of photocatalytic performance of PVA/NCD-TiO$_2$ film can be proposed: under visible light illumination, the degradation of organic contaminant is caused by direct oxidation from photo-generated holes (major) and indirect oxidation by O$_2^{\bullet-}$ (minor); more photogenerated holes receivers due to doping with the impurities (i.e., N and C) and more photogenerated electron receivers due to the formation of oxygen vacancy can contribute to the inhibited recombination of photogenerated electrons and holes (as proven by photoluminescence spectroscopy analysis [34]) and benefit the high photocatalytic activity of the catalyst; PVA functions to provide transparent support for the NCD-TiO$_2$ nanoparticle and to reduce the cost of the subsequent treatment processes for water, facilitating the practical application of visible-light-activated TiO$_2$ nanomaterial (e.g., NCD-TiO$_2$).

### 2.3. Byproducts of the Photodegradation of LEV Catalyzed by the PVA/NCD-TiO₂ Film

Mass spectrometric analysis of the intermediates of LEV during photocatalytic degradation was performed in this work. Seventeen byproducts were deduced by referring to previous research [47–51], and the reaction mechanism derived (i.e., mainly photogenerated holes and partially superoxide radical) was in line with our previous work [34]. The detailed information of the deduced byproducts is listed in Table 2. In the comparison of the structures of the byproducts with those of LEV, the piperazinyl group of LEV appears transformed in all byproducts. Among the 17 byproducts, only 2 byproducts (P11 and P14) have changes in both the quinolone ring and the piperazinyl group, while the remaining 15 byproducts only have the change in the piperazinyl group.

**Table 2.** Proposed byproducts in the process of photocatalytic removal of LEV.

| Name | m/z | Retention Time (min) | Molecular Weight | Molecular Formula | Proposed Structure |
|------|-----|----------------------|------------------|-------------------|--------------------|
| LEV | 362 | 19.8 | 361 | $C_{18}H_{20}FN_3O_4$ | |
| P1 | 378 | 20.6 | 377 | $C_{18}H_{20}FN_3O_5$ | |
| P2 | 376 | 23.9 | 375 | $C_{18}H_{18}FN_3O_5$ | |
| P3a | 390 | 20.8/21.4 | 389 | $C_{18}H_{16}FN_3O_6$ | |
| P3b | 390 | 20.8/21.4 | 389 | $C_{18}H_{16}FN_3O_6$ | |
| P4 | 392 | 21.8 | 391 | $C_{18}H_{18}FN_3O_6$ | |
| P5a | 364 | 19.0/23.0 | 363 | $C_{17}H_{18}FN_3O_5$ | |
| P5b | 364 | 19.0/23.0 | 363 | $C_{17}H_{18}FN_3O_5$ | |
| P6 | 336 | 19.6 | 335 | $C_{16}H_{18}FN_3O_4$ | |
| P7a | 350 | 18.9/22.2 | 349 | $C_{16}H_{16}FN_3O_5$ | |
| P7b | 350 | 18.9/22.2 | 349 | $C_{16}H_{16}FN_3O_5$ | |

**Table 2.** *Cont.*

| Name | m/z | Retention Time (min) | Molecular Weight | Molecular Formula | Proposed Structure |
|---|---|---|---|---|---|
| P8 | 322 | 19.5 | 321 | $C_{15}H_{16}FN_3O_4$ |  |
| P9 | 307 | 21.3 | 306 | $C_{14}H_{11}FN_2O_5$ |  |
| P10 | 279 | 22.7 | 278 | $C_{13}H_{11}FN_2O_4$ |  |
| P11 | 368 | 17.0 | 367 | $C_{16}H_{18}FN_3O_6$ |  |
| P12 | 376 | 23.9 | 375 | $C_{18}H_{18}FN_3O_5$ |  |
| P13 | 348 | 19.8 | 347 | $C_{17}H_{18}FN_3O_4$ |  |
| P14 | 324 | 17.0 | 323 | $C_{15}H_{18}FN_3O_4$ |  |

According to the structural information of the byproducts and the photocatalytic reaction mechanism derived previously, the possible reaction pathways for the photocatalytic degradation of LEV are proposed including (i) the ring opening of piperazinyl group and (ii) the ring opening of quinolone core.

In path (i), as shown in Figure 8, the piperazinyl group of LEV can be attacked by the oxidizing reactive species (i.e., holes and/or superoxide radicals) and hydroxylated to produce the first byproduct P1. The generated hydroxyl group of P1 can then be further oxidized to a ketone byproduct P2. Two possible further oxidations are proposed for P2: (1) the methyl group on the side chain of alkylamine of P2 is oxidized to an aldehyde P3a, and (2) the piperazine ring of P2 can be continuously oxidized to produce a diketone P3b which is relatively stable since the para ketone is on the piperazine ring. If the formed diketone is an ortho diketone, which is extremely unstable, the C–C bond between the ketones would rapidly disconnect, i.e., the ring opening of the piperazine ring, to generate P4 with a dialdehyde structure [52]. P5a and P5b can be generated by oxidizing and shedding one aldehyde group from byproduct P4, while shedding the other aldehyde group can generate P6. Subsequently, the intermediate product P6 may undergo two pathways: (1) the methyl group on the alkylamine is oxidized to an aldehyde to generate P7a, where the aldehyde is further oxidized to shed and generate P8; (2) the –$CH_2$– next to the aniline on the hemipiperazinyl of P6 is ketonized to produce byproduct P7b, which would result in the C-C bond cleavage and the generation of byproduct P9. The aldehyde group on the aniline of P9 is further oxidized and shed into P10. The proposed mineralization path of LEV to the product P10 presents the whole process in which the piperazinyl group of LEV is oxidized off in steps.

**Figure 8.** Proposed pathways (i) of the degradation of LEV.

In path (ii), as shown in Figure 9, during the visible-light photocatalytic degradation of LEV, two byproducts, P11 and P14, can be the generated, with the quinolone ring being involved in the reaction. Byproduct P11 is generated by the decarboxylation of the quinolone ring of the byproduct P1 followed by ring opening. The methyl group on the alkylamine side chain of LEV can be first oxidized to generate P12. Then, the aldehyde group of the byproduct P12 is oxidized and shed to produce byproduct P13. Finally, the byproduct P14 can be generated from the product P13 via decarboxylation ring opening. The elimination of the quinolone ring of the contaminants (e.g., P11 and P14) indicates the reduced antibacterial activity [53].

**Figure 9.** Proposed pathways (ii) of the degradation of LEV.

## 3. Materials and Methods

### 3.1. Materials

PVA with an average degree of polymerization of 1799 with a hydrolysis degree of 98.0~99.0% was purchased from Shanghai Maclin Biochemical Technology Co., Ltd. (Shanghai, China). NCD-TiO$_2$ was prepared with our previously reported approach [34]. Phenol (99%) was purchased from Macklin Biochemical Co. (Shanghai, China), and LEV (>98%) was purchased from Shanghai Aladdin Biochemical Technology Co., Ltd. (Shanghai, China). All chemicals were applied without further purification. Deionized water was used to prepare experimental solutions.

### 3.2. Preparation of PVA/NCD-TiO$_2$ Nanocomposites

PVA/NCD-TiO$_2$ hybrid membranes were prepared through solution casting and heat-treatment process. NCD-TiO$_2$ nanoparticles (0.2 g) were dispersed in 100 mL of deionized water with ultrasonication, after which 2.5 g of polyvinyl alcohol (PVA1799) was immediately added into the mixture with continuous magnetic stirring at 95 °C for

1 h. The mixture was then cooled to 60 °C and stirred continuously for 3 h. After that, the mixture was poured into PTFE molds 0.5 cm in thickness at room temperature for 48 h to eliminate the bubbles. Then, the resulting sample underwent thermal treatment at certain temperatures (i.e., 80 °C, 120 °C, 160 °C) in a vacuum. The membranes were washed with deionized water and dried before use. In this paper, the samples are named by the preparation approach: X-T, where X indicates the weight ratio of $NCD-TiO_2$ to PVA (i.e., 4 wt%, 8 wt%, 12 wt%, 16 wt%, 20 wt%), and T represents the treatment temperature (i.e., 0, 80, 120, 160 in °C). For example, $8wt\%-120-PVA/NCD-TiO_2$ indicates that the weight ratio of $NCD-TiO_2$ to PVA is 8 wt% and the treatment temperature is 120 °C. (Through ICP analysis, the $8\ wt\%-120-PVA/NCD-TiO_2$ sample contains $TiO_2$ around $5.9 \pm 0.6$ wt%.) The sample of pure PVA was prepared with the same procedure but without the addition of $NCD-TiO_2$.

### 3.3. Characterization

The morphology of the hybrid films was observed under field emission scanning of an electron microscope (Fei quanta 650, Hillsboro, OR, USA) at an accelerating voltage of 5 kV. X-ray diffraction (XRD) patterns of different phases of the $PVA/NCD-TiO_2$ nanocomposite films were investigated using an X-ray diffractometer (Science Direct Ultima IV, Akishima, Japan) with Cu K$\alpha$ ($\lambda = 1.54$ Å) in a Bragg angle ($2\theta$) range of $10° \le 2\theta \le 80°$. The shape and particle size of the $NCD-TiO_2$ nanoparticles were inspected by transmission electron microscope (TEM, a Tecnai G20, FEI, Amsterdam, The Netherlands) with the applied voltage being 200 KV and the gun-type being LaB6. The chemical interaction between PVA and $NCD-TiO_2$ nanoparticles was confirmed using attenuated total reflection/Fourier transform infrared spectroscopy (ATR/FTIR, Thermo Scientific Nicolet 10, Waltham, MA, USA).

The intermediates analysis was performed with a HPLC/MS system (Agilent 1200/6200 Ion Trap LC/MS, Palo Alto, CA, USA) equipped with an X Bridge C18 column (250 mm $\times$ 4.6 mm, 5 $\mu$m particle size). The mobile phase was composed of methanol (A), acetonitrile (B), and 1% (*v/v*) formic acid (98%, Aladdin, Shanghai, China) in ultrapure water (C, pH = 3.0, adjusted by adding ammonium formate), and the flow rate was 1 mL/min. The composition of the mobile phase throughout analysis was as follows: 2% of A, 13% of B, and 95% of C for 30 min; then 57% of A, 38% of B, and 5% of C for 10 min; and 2% of A, 13% of B, and 95% of C for the last 20 min. The injection volume was 20 $\mu$L and the column temperature was 30 °C. MS was performed through operating in the positive ion mode using ESI, while the capillary was +4.0 KeV and the temperature of drying gas was 350 °C. MS was scanned with a mass range from *m/z* 50 to 700.

### 3.4. Photocatalytic Degradation

In order to investigate the photocatalytic activities of different $PVA/NCD-TiO_2$ films under visible-light irradiation, the experiments for the destruction of various contaminants, including phenol and LEV, were conducted under visible light in a self-made experimental installation. The reactor was a glass Petri dish (100 mm (ø) $\times$ 50 mm (h)), where the stock solution of the contaminant was spiked in ultrapure water (pH = 5.7) to form the reaction solution with the initial concentration of phenol or LEV being 20 $\mu$M. The different $PVA/NCD-TiO_2$ films were added into the reactor with continuous stirring. The final volume of the reaction solution was 50.0 mL. The reactor was sealed, including its glass cover, and cooled with fans during the reaction. Four fluorescent lamps (8 W, Philips) with a UV light filter (JB420, Yongxing Sensing, Beijing, China) were employed as the source of visible light whose intensity was measured to be 11.58 mW/cm$^2$ using a radiant power meter (FZ-A, Photoelectric Instrument Factory of BNU). The surface of the solution in the reactor was 15 cm away from the lamps. During the photocatalytic process, 2 mL samples were taken after 0, 30, 60, 90, and 120 min and filtered with nylon syringe filters (0.45 $\mu$m).

The concentration of each contaminant in the samples obtained at different irradiation time intervals was quantified using a high-performance liquid chromatograph (HPLC, e2695, Waters, Milford, MA, USA) with a 2489 UV/VIS detector set at 270 nm

and 296 nm for phenol and LEV measurement, respectively. An XBridge C18 column (250 mm $\times$ 4.6 mm, 5 µm particle size) served as the stationary phase. For phenol measurement, a mixture of ultrapure water and acetonitrile (99.9%, Aladdin, Shanghai, China) in a volume ratio of 50:50 was employed as the mobile phase with the column temperature, flow rate, and injection volume being 30 °C, 1 mL/min, and 20 µL, respectively. For LEV HPLC measurement, the mobile phase included three parts: acetonitrile (A), methanol (B, 99.9%, Aladdin, Shanghai, China), and 1% (*v/v*) formic acid (98%, Aladdin, Shanghai, China) in ultrapure water (C, pH = 3.0, adjusted by adding Ammonium formate), in which the A:B:C were in a volume ratio of 3:2:15. The column temperature was 35 °C, the flow rate was 1 mL/min, and the injection volume was 50 µL. The PVA/NCD-TiO$_2$ films have also been studied for their recyclability. After each photocatalytic reaction, the films were washed several times with deionized water to eliminate any residual organics and dried for further use in the next cycle.

## 4. Conclusions

The as-prepared PVA/NCD-TiO$_2$ film exhibited stable photocatalysis in degrading phenol and LEV, with approximately 40% and 42% removal for 2 h (8 wt%-120-PVA/NCD-TiO$_2$). PVA/NCD-TiO$_2$ film showed stable photocatalysis and multiple cycle stability in degrading LEV after thermal treatment at 120 °C. The low cost PVA/NCD-TiO$_2$ film is recyclable and has a promising application in degrading pharmaceutical wastewater such as LEV. The main intermediates and possible LEV degradation pathways were proposed based on LC–MS results. This study inspires the design of high stability and excellent photocatalyst for practical application.

**Supplementary Materials:** The following supporting information can be downloaded at: https://www.mdpi.com/article/10.3390/catal12111336/s1, Figure S1: SEM images of the surfaces of 8wt%-80-PVA/NCD-TiO$_2$; Figure S2: The Raman spectra of different synthesized materials; Figure S3: The UV–vis absorption spectra of different synthesized materials.

**Author Contributions:** Conceptualization, G.Z.; methodology, G.Z.; software, A.J.; validation, W.Y.; formal analysis, A.J.; investigation, A.J.; resources, A.J.; data curation, A.J.; writing—original draft preparation, A.J.; writing—review and editing, G.Z.; visualization, W.Y.; supervision, X.H.; project administration, X.H.; funding acquisition, X.H. All authors have read and agreed to the published version of the manuscript.

**Funding:** This work was funded by Zhejiang Provincial Natural Science Foundation of China (grant number LQ19B070005) and Zhejiang Garden Pharmaceutical Corporation Limited (contract no. 2017330004001542 and project no. KYY-HX-20170062).

**Institutional Review Board Statement:** Not applicable.

**Conflicts of Interest:** The authors declare no conflict of interest.

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
