# Peer review of "A Study of the Degradation of LEV by Transparent PVA/NCD-TiO2 Nanocomposite Films with Enhanced Visible-Light Photocatalytic Activity"

_catalysts, doi:10.3390/catal12111336_

Round 1

Reviewer 1 Report

In the article, the authors synthesized PVA/NCD-TiO2 (NCD for nitrogen and carbon co-doped titanium dioxide 14 nano-catalyst) film to degrade antibiotic such as levofloxacin in water by semiconductor photocatalysis. To mix PVA and NCD-TiO2 and make nano-composite, which exhibited high efficiency, reusability and stability in photocatalysis. The main intermediates and possible LEV degradation pathways were analyzed based on LC–MS results. The as-prepared film is recyclable and has a promising application in degrading pharmaceutical wastewater. It should be noted that doped TiO2 is well-studied in recent years, and in the manuscript, some information of the photocatalyst should be supplied or clarified. Moreover, the manuscript provides abundant analysis data of products, verifies the material in natural water, and presents the possible reaction path. The reviewer suggests the manuscript a major revision to focus more on the catalysis.

Some problems and suggestions are as follows, which could help authors to improve the quality of the work.

1.    In page 3, line 84, the name of “8wt%-0-PVA/TiO2” and “8wt%-120-PVA/TiO2” should be clarified and highlighted, especially the “0” and “120”, what’s the meaning of the two numbers?

2.    According to the manuscript, is TiO2 in “8wt%-0-PVA/TiO2” and “8wt%-120-PVA/TiO2” refers to NCD-TiO2, not pure TiO2? If NCD-TiO2 is used, the author should modify the name to avoid confusion.

3.    In page 7, from line 219 to line 221, authors explained that temperature influence hydrogen bonds between the O-H in PVA and the terminal hydroxyl groups on the TiO2 surface. However, it is still confusing how temperature influence photocatalytic efficiency of prepared photocatalysts.

4.    As a kind of polymer, could PVA influence the photocatalytic efficiency of NCD-TiO2? The author should supply the photocatalytic data of bare NCD-TiO2.

5.    For the same reason in suggestion 4, The author should provide UV-Vis spectra of synthesized PVA/NCD-TiO2 film and bare NCD-TiO2.

6.    Levofloxacin is a synthetic fluoroquinolone antibiotic, which means levofloxacin contains fluorine elements. Therefore, the products and byproducts from the photocatalytic process have fluorine elements as well. Are these products and byproducts toxic in water?

Author Response

Response to Reviewer 1 Comments

In the article, the authors synthesized PVA/NCD-TiO2 (NCD for nitrogen and carbon co-doped titanium dioxide 14 nano-catalyst) film to degrade antibiotic such as levofloxacin in water by semiconductor photocatalysis. To mix PVA and NCD-TiO2 and make nano-composite, which exhibited high efficiency, reusability and stability in photocatalysis. The main intermediates and possible LEV degradation pathways were analyzed based on LC–MS results. The as-prepared film is recyclable and has a promising application in degrading pharmaceutical wastewater. It should be noted that doped TiO2 is well-studied in recent years, and in the manuscript, some information of the photocatalyst should be supplied or clarified. Moreover, the manuscript provides abundant analysis data of products, verifies the material in natural water, and presents the possible reaction path. The reviewer suggests the manuscript a major revision to focus more on the catalysis.

Some problems and suggestions are as follows, which could help authors to improve the quality of the work.

Point 1: In page 3, line 84, the name of “8wt%-0-PVA/TiO2” and “8wt%-120-PVA/TiO2” should be clarified and highlighted, especially the “0” and “120”, what’s the meaning of the two numbers?

Response 1: We have clarified the name of the samples in 3.2. Preparation of PVA/NCD-TiO2 nanocomposites. Besides, we have modified the name of the sample from “…PVA/TiO2” to “…PVA/ NCD-TiO2” to avoid confusion in our revised manuscript.

Point 2: According to the manuscript, is TiO2 in “8wt%-0-PVA/TiO2” and “8wt%-120-PVA/TiO2” refers to NCD-TiO2, not pure TiO2? If NCD-TiO2 is used, the author should modify the name to avoid confusion.

Response 2: We thank the reviewer very much for the comment. Yes, the samples refer to NCD-TiO2. In our revised manuscript, we have modified the name of the sample from “…PVA/TiO2” to “…PVA/ NCD-TiO2” to avoid confusion, as suggested.

Point 3: In page 7, from line 219 to line 221, authors explained that temperature influence hydrogen bonds between the O-H in PVA and the terminal hydroxyl groups on the TiO2 surface. However, it is still confusing how temperature influence photocatalytic efficiency of prepared photocatalysts.

Response 3: The hydrogen bonds between the O-H in PVA and the terminal hydroxyl groups on the TiO2 surface are essential for the interaction between NCD-TiO2 and PVA, which can avoid the loss of catalyst and maintain the performance of catalyst during cyclic usage. In order to clarify the function of hydrogen bonds, the last sentence of this paragraph has been modified into “In summary, the heating treatment requires suitable temperature, too high or too low temperature would not form hydrogen bonds between the O-H in PVA and the terminal hydroxyl groups on the TiO2 surface, which would eventually affect the performance of catalyst during cyclic usage.” in the revised manuscript.

Point 4: As a kind of polymer, could PVA influence the photocatalytic efficiency of NCD-TiO2? The author should supply the photocatalytic data of bare NCD-TiO2.

Response 4: The addition of PVA would decrease the photocatalytic efficiency of NCD-TiO2 to some extent due to the sacrifice of active surface of NCD-TiO2 when the particles are immobilized in PVA. The immobilization would benefit subsequence treatment processes and reduce the total cost for water treatment. The photocatalytic data of bare NCD-TiO2 has been published previously as “Huang, X.W.; Yang, W.Q.; Zhang, G.; Yan, L.; Zhang, Y.C.; Jiang, A.H.; Xu, H.L.; Zhou, M.; Liu, Z.J.; Tang, H.D. Alternative synthesis of nitrogen and carbon co-doped TiO2 for removing fluoroquinolone antibiotics in water under visible light. Catal. Today 2021, 361, 11–16”. And we have added the sentence “The photocatalytic data of NCD-TiO2 can be found in our previous work [33].” in 2.2.1 to clarify.

Point 5: For the same reason in suggestion 4, The author should provide UV-Vis spectra of synthesized PVA/NCD-TiO2 film and bare NCD-TiO2.

Response 5: The UV-Vis spectra of synthesized PVA/NCD-TiO2 film and bare NCD-TiO2 have been provided in Fig. S1 in Supplementary Material and discussed in 2.2.1 “The UV–vis absorption spectrum of PVA/NCD-TiO2 (see Fig. S1 in Supplementary Information (SI)) show that the synthesized membrane still has visible-light activity hinting its photocatalytic performance.” as suggested.

Point 6: Levofloxacin is a synthetic fluoroquinolone antibiotic, which means levofloxacin contains fluorine elements. Therefore, the products and byproducts from the photocatalytic process have fluorine elements as well. Are these products and byproducts toxic in water?

Response 6: We thank the reviewer very much for the comment. After reviewing the references, we found that the ability of fluoroquinolones to inhibit bacterial activity is linked to the quinolone moiety of the molecule, which is the key issue that we concern. Hence, the elimination of quinolone moiety of the containments and their byproducts would be important in water treatment. Therefore, we added a sentence “The elimination of quinolone ring of the containments (e.g. P11 and P14) indicates the reduced antibacterial activity.” in last paragraph in 2.3 to clarify.

Reviewer 2 Report

The authors present the manuscript entitled A Study of the Degradation of LEV by Transparent PVA/NCD-2 TiO2 Nanocomposite Films with Enhanced Visible-light Photo-3 catalytic Activity.

It is recommended that the following observations and recommendations be addressed in order to improve this manuscript:

• Analyze by HR-TEM all the synthesized samples and make a comparison between the microscopies obtained.

• Label the crystallographic planes in the XRD Figure.

• Detail the synthesis of all materials

• The materials were calcined?

• Explain the role of the OH, C-H, C-C and C-O groups in the photocatalytic mechanism.

• If the above groups are important, work the ATR/FTIR results in absorbance mode and perform deconvolutions to better visualize these results. Review the following bibliography: Tzompantzi, F., Castillo-Rodríguez, J. C., et al (2022 ). Facile synthesis of ZrO2-Bi2O2 (CO) 3 composite materials prepared in one-pot synthesis for high photoactivity in efficient hydrogen production. Journal of Photochemistry and Photobiology A: Chemistry, 423, 113594.

• Maintain the color code in all graphs.

• Delve into the characterization of materials with UV-Vis diffuse reflectance spectroscopy, photoluminescence spectroscopy and electrochemical techniques.

• With the results of all the above, propose a photocatalytic reaction mechanism where the role of the best material is observed, as well as the generation of the electron-hole pair and the degradation of the molecule.

Author Response

Response to Reviewer 2 Comments

The authors present the manuscript entitled A Study of the Degradation of LEV by Transparent PVA/NCD-2 TiO2 Nanocomposite Films with Enhanced Visible-light Photo-3 catalytic Activity.

It is recommended that the following observations and recommendations be addressed in order to improve this manuscript:

Point 1: Analyze by HR-TEM all the synthesized samples and make a comparison between the microscopies obtained.

Response 1: Because the membrane is macroscopic, it’s more convenient to observe it with SEM. And the NCD-TiO2 nanoparticles are microscopic and observed with TEM. We have shown the SEM images of pure PVA, 8wt%-0-PVA/NCD-TiO2, 8wt%-120-PVA/NCD-TiO2, and 8wt%-160-PVA/NCD-TiO2 in Figure 1 and discussed the comparison in 2.1.1.

Point 2: Label the crystallographic planes in the XRD Figure.

Response 2: We have labeled the XRD Figure (Figure 2) with crystallographic planes as suggested.

Point 3: Detail the synthesis of all materials

Response 3: The synthesis of the materials are described in 3.2 and more details have been added as suggested.

Point 4: The materials were calcined?

Response 4: The NCD-TiO2 nanoparticles were synthesized through calcination at 430 °C, while the PVA/NCD-TiO2 films were synthesized through low temperature heat treatment (0-160 °C) under vacuum.

Point 5: Explain the role of the OH, C-H, C-C and C-O groups in the photocatalytic mechanism.

Response 5: The interaction between PVA and NCD-TiO2 can be characterized through ATR/FT-IR analysis. Stronger interaction between PVA and NCD-TiO2 (i.e. higher content of these groups (e.g. hydroxyl, C-H, and C-O)) can facilitate PVA to be transparent support for TiO2, which would eventually benefit the photocatalytic treatment for water purification.

Point 6: If the above groups are important, work the ATR/FTIR results in absorbance mode and perform deconvolutions to better visualize these results. Review the following bibliography: Tzompantzi, F., Castillo-Rodríguez, J. C., et al (2022 ). Facile synthesis of ZrO2-Bi2O2 (CO) 3 composite materials prepared in one-pot synthesis for high photoactivity in efficient hydrogen production. Journal of Photochemistry and Photobiology A: Chemistry, 423, 113594.

Response 6: We thank the reviewer very much for the comment. In order to better explore the internal molecular structure changes of materials under different heating treatment in this work, the infrared spectra of the samples were investigated by applying the heating temperature as the variable to form a data set for two-dimensional correlation spectral analysis. Two dimensional correlation analysis provided two different maps: the synchronous spectrum shows the homology of the spectrum signal (the change direction and change rate of functional groups), and the asynchronous correlation spectrum correlates the information with the sequence of events (i.e. the change sequence of thermal decomposition/synthesis of functional groups). Besides, in our revised manuscript, the bibliography (Tzompantzi, F., Castillo-Rodríguez, J. C., et al (2022). Facile synthesis of ZrO2-Bi2O2 (CO) 3 composite materials prepared in one-pot synthesis for high photoactivity in efficient hydrogen production. Journal of Photochemistry and Photobiology A: Chemistry, 423, 113594.) is referred.

Point 7: Maintain the color code in all graphs.

Response 7: The color code is uniformed as suggested.

Point 8: Delve into the characterization of materials with UV-Vis diffuse reflectance spectroscopy, photoluminescence spectroscopy and electrochemical techniques.

Response 8: The UV-Vis spectra of synthesized PVA/NCD-TiO2 film and bare NCD-TiO2 have been provided in Fig. S1 in Supplementary Material and discussed in 2.2.1 “The UV–vis absorption spectrum of PVA/NCD-TiO2 (see Fig. S1 in Supplementary Information (SI)) show that the synthesized membrane still has visible light activity hinting its photocatalytic performance.” as suggested. The photoluminescence spectroscopy of NCD-TiO2 has been investigated in our previous work (“Huang, X.W.; Yang, W.Q.; Zhang, G.; Yan, L.; Zhang, Y.C.; Jiang, A.H.; Xu, H.L.; Zhou, M.; Liu, Z.J.; Tang, H.D. Alternative synthesis of nitrogen and carbon co-doped TiO2 for removing fluoroquinolone antibiotics in water under visible light. Catal. Today 2021, 361, 11–16”.) and shows a slower electron-hole recombination of NCD-TiO2 sample comparing normal TiO2, indicating the doped TiO2 would have inhibition of electron-hole recombination and eventually better photocatalytic performance. The main function of PVA is providing transparent support for NCD-TiO2 and benefiting subsequence treatment processes by reducing the total cost for water treatment.

Point 9: With the results of all the above, propose a photocatalytic reaction mechanism where the role of the best material is observed, as well as the generation of the electron-hole pair and the degradation of the molecule.

Response 9: We thank the reviewer very much for the comment. In our previous work regarding NCD-TiO2 nanocatalyst, we have proposed that the photocatalytic degradation of LEV under visible light illumination is mainly through direct oxidation by photo-generated holes in the valence band of NCD-TiO2, and at a minor extent through indirect oxidation by O2•−, which is transformed from the reduction of molecular oxygen by electron generated in conduction band of TiO2. On one hand, the formation of mid-gaps or color centers in the TiO2 due to doping with the impurities (i.e. N and C) would largely decrease the effective bandgap of catalyst and hence decrease the activation energy required in the photocatalytic reaction; on the other hand, the formation of oxygen vacancy on the surface of TiO2 could facilitate Ti4+ to trap the photogenerated electron to form Ti3+, while the surface doped species could receive the photogenerated holes from valence band and contribute to the degradation of pollutants, and hence the recombination of photogenerated electrons and holes can be efficiently inhibited. The above mentioned two aspects would eventually contribute to the high visible-light activity of NCD-TiO2. And the main function of PVA is providing transparent support for NCD-TiO2 nanoparticle and reducing the cost of subsequence treatment processes for water treatment.

Reviewer 3 Report

In this paper, the authors studies the degradation of LEV and phenol by a PVA/NCD-TiO2 nanocomposite films. I believe that the results shown in this paper should be better explained and contrasted. So, I believe that major revision must be made to the manuscript to be suitable for publication.

Figure 1 shows several SEM and TEM images, but I do not understand if figures c and d correspond to the PVA/NCD-TiO2 material or to a different one, if we refer to the same sample, they should not be named as: 8wt%- 0-PVA/NCD-TiO2.

The conclusion shown by FTIR in which the incorporation of TiO2 into PVA is demonstrated is not clear, another technique should be used to corroborate the information, for example, the use of Raman spectroscopy would be necessary to support the results.

Fig 5 shows the degradation of the different contaminants in contact with the materials, very different effects are produced between them, which are not clearly explained in the manuscript, in turn, fig 5b, c and d are not detailed in the manuscript. I think these results should be better explained.

How has the amount of TiO2 been calculated? Techniques such as XRF or ICP-MS should be performed. This would help to understand the conclusions observed in section 2.2.2.

Have blank tests been carried out without a photocatalyst? Is there degradation of the contaminants due to contact with light?

It would be necessary to carry out a characterization after the photocatalysis process, observing if leaching occurs in the materials.

On the line 320 there is a grammatical error, where it indicates "altrasonication" should be "ultrasonication"

Author Response

Response to Reviewer 3 Comments

In this paper, the authors studies the degradation of LEV and phenol by a PVA/NCD-TiO2 nanocomposite films. I believe that the results shown in this paper should be better explained and contrasted. So, I believe that major revision must be made to the manuscript to be suitable for publication.

Point 1: Figure 1 shows several SEM and TEM images, but I do not understand if figures c and d correspond to the PVA/NCD-TiO2 material or to a different one, if we refer to the same sample, they should not be named as: 8wt%- 0-PVA/NCD-TiO2.

Response 1: We have changed the figure caption of Figure 1 to be “Figure 1. SEM images of the surfaces of (a) pure PVA, (b) 8wt%-0-PVA/NCD-TiO2, (c) 8wt%-120-PVA/NCD-TiO2, (d) 8wt%-160-PVA/NCD-TiO2; TEM images of (e)-(f) NCD-TiO2 nanoparticle.” to clarify and avoid confusion, as suggested.

Point 2: The conclusion shown by FTIR in which the incorporation of TiO2 into PVA is demonstrated is not clear, another technique should be used to corroborate the information, for example, the use of Raman spectroscopy would be necessary to support the results.

Response 2: Raman spectroscopy has been applied (see below), as suggested. The Raman spectrum of synthesized PVA/NCD-TiO2 shows a combination of the spectra of PVA and NCD-TiO2. We also reviewed other relevant literatures and found FTIR or ATR/FT-IR are the commonest and reliable technique to support the interaction between PVA and NCD-TiO2.

Point 3: Fig 5 shows the degradation of the different contaminants in contact with the materials, very different effects are produced between them, which are not clearly explained in the manuscript, in turn, fig 5b, c and d are not detailed in the manuscript. I think these results should be better explained.

Response 3: We thank the reviewer very much for the comment. We made a mistake in Figure 5: (a) and (b) should be the results of phenol degradation, while (c) and (d) are those of LEV degradation (NOT (a) and (c) for phenol, and (b) and (d) for LEV). Hence the different contaminants actually show similar effect when contact with the materials: the PVA/NCD-TiO2 film without thermal treatment (i.e. 8wt%-0-PVA/NCD-TiO2) or low temperature treatment (i.e. 8wt%-80-PVA/NCD-TiO2) has better initial photocatalytic activities, however, its later photocatalytic activity shows a significant decrease; the catalytic efficiency of 8wt%-120-PVA/NCD-TiO2 only showed slight decrease within four cycles, whereas the 8wt%-160-PVA/NCD-TiO2 film showed a lower catalytic activity.

Point 4: How has the amount of TiO2 been calculated? Techniques such as XRF or ICP-MS should be performed. This would help to understand the conclusions observed in section 2.2.2.

Response 4: ICP analysis has been applied to obtain the amount of TiO2 to be around 5.9 (±10%) wt% in the synthesized film (8wt%-120-PVA/NCD-TiO2), as suggested.

Point 5: Have blank tests been carried out without a photocatalyst? Is there degradation of the contaminants due to contact with light?

Response 5: The degradation of LEV through direct photolysis has been tested and the degradation is negligible without photocatalyst.

Point 6: It would be necessary to carry out a characterization after the photocatalysis process, observing if leaching occurs in the materials.

Response 6: We thank the reviewer very much for the comment. The ICP analysis has been carried out for the sample (8wt%-120-PVA/NCD-TiO2) before and after photocatalysis, as suggested. The leaching of TiO2 is within 10%.

Point 7: On the line 320 there is a grammatical error, where it indicates "altrasonication" should be "ultrasonication"

Response 7: The "altrasonication" has been replaced by "ultrasonication" as suggested.

Round 2

Reviewer 1 Report

The revision is reasonable. It is recommended to be accepted. 

Author Response

We appreciate the suggestions and review of the reviewer.

Reviewer 2 Report

The authors present the manuscript entitled A Study of the Degradation of LEV by Transparent PVA/NCD-2 TiO2 Nanocomposite Films with Enhanced Visible-light Photo-3 catalytic Activity.

The authors addressed some of the comments. address the following recommendations to improve this manuscript:

- Analyze by HR-TEM all the synthesized samples and perform a comparison between the obtained microscopies.

- To detail the synthesis of all the materials.

- To deepen the characterization of the materials with UV-Vis diffuse reflectance spectroscopy and photoluminescence spectroscopy.

- Based on their results , propose a photocatalytic reaction mechanism where the role of the best material is observed, as well as the generation of the electron-hole pair and the degradation of the molecule.

After addressing the recommendations and observations, it is recommended to accept this article for publication.

The authors attended 35% of the recommendations, it is suggested that they consider the recommendations and then the article is accepted.

Reviewer 3 Report

In this paper, the authors studies the degradation of LEV and phenol by a PVA/NCD-TiO2 nanocomposite films. The authors use different techniques for its determination. Also, the authors have responsed to all my questions and they have greatly improved the paper. Although, some of the information explained in your answer could be included in the manuscript

Raman results should at least be included in the supplementary material.

The catalytic test without catalyst must be included.

The result obtained in the leaching test must be included.

After looking at the paper carefully, I consider that this paper need a minor revision for publication.

Author Response

Response to Reviewer 3 Comments

In this paper, the authors studies the degradation of LEV and phenol by a PVA/NCD-TiO2 nanocomposite films. The authors use different techniques for its determination. Also, the authors have responsed to all my questions and they have greatly improved the paper. Although, some of the information explained in your answer could be included in the manuscript

Point 1: Raman results should at least be included in the supplementary material.

Response 1: We have included in the supplementary material and added 2.1.2 “The Raman spectra (see Fig. S1 in Supplementary Information (SI)) also support the conclusions.”, as suggested.

Point 2: The catalytic test without catalyst must be included.

Response 2: We have added the result of photolytic test without catalyst in the section 2.2.1 “Besides, the direct photolysis of phenol and LEV under visible light is measured to be negligible.”, as suggested.

Point 3: The result obtained in the leaching test must be included.

Response 3: We have included the results from leaching test in section 2.2.1 “In addition, the leaching of TiO2 after photocatalysis were measured through ICP analysis to be within 10% for 8wt%-120-PVA/NCD-TiO2.”, as suggested.

After looking at the paper carefully, I consider that this paper need a minor revision for publication.
